# GENERATIVE MODEL BASED NOISE ROBUST TRAINING FOR UNSUPERVISED DOMAIN ADAPTATION

## ABSTRACT

Target domain pseudo-labelling has shown effectiveness in unsupervised domain adaptation (UDA). However, pseudo-labels of unlabeled target domain data are inevitably noisy due to the distribution shift between source and target domains. This paper proposes a generative model-based noise-robust training method (GeMo-NoRT), which eliminates domain shift while mitigating label noise. GeMo-NoRT incorporates a distribution-based class-wise feature augmentation (D-CFA) and a generative-discriminative classifier consistency (GDC), both based on the class-wise target distributions modelled by generative models. D-CFA minimizes the domain gap by augmenting the source data with distribution-sampled target features, and trains a noise-robust discriminative classifier by using target domain knowledge from the generative models. GDC regards all the class-wise generative models as generative classifiers and enforces a consistency regularization between the generative and discriminative classifiers. It exploits an ensemble of target knowledge from all the generative models to train a noise-robust discriminative classifier and eventually gets theoretically linked to the Ben-David domain adaptation theorem for reducing the domain gap. Extensive experiments on Office-Home, PACS, and Digit-Five show that our GeMo-NoRT achieves state-of-the-art under single-source and multi-source UDA settings.

## 1 INTRODUCTION

Convolutional neural networks (CNNs) trained by large amounts of training data have achieved remarkable success on a variety of computer vision tasks (Simonyan & Zisserman, 2014; Szegedy et al., 2015; He et al., 2016; Long et al., 2015a; He et al., 2019). However, when a well-trained CNN model is deployed in a new environment, its performance usually degrades drastically. This is because the test data (of the target domain) is typically from a different distribution from the training data (of source domains). Such distribution mismatch is also known as domain gap. A popular solution to tackling the domain gap issue is unsupervised domain adaptation (UDA) (Gretton et al., 2012; Long et al., 2015b; 2016; Tzeng et al., 2014; Balaji et al., 2019; Xu et al., 2018).

UDA can be divided into two main sub-settings: single-source domain adaptation (SSDA) and multi-source domain adaptation (MSDA), according to the number of source domains. Early works have mainly focused on the single-source scenarios (Gretton et al., 2012; Long et al., 2015b; Ganin et al., 2016; Tzeng et al., 2017). Nevertheless, in real-world applications, the source domain data can be collected from various deployment environments, leading to multiple source setting. MSDA thus has been receiving more attention recently. Most UDA methods, including both SSDA and MSDA, tried to reduce domain gap by domain distribution alignment (Zhao et al., 2018; Xu et al., 2018; Peng et al., 2019; Li et al., 2021c). Latest methods (Wang et al., 2020; Li et al., 2021a) further utilized class information for class-wise alignment, with pseudo-labels used for unlabeled target data. These methods have shown effectiveness for UDA. However, due to the domain gap, a model trained on the source domains cannot correctly classify all the target instances, leading to target domain pseudo-labels inevitably being noisy. If these noisy labels are directly used as supervision, their negative impact can be amplified or accumulated through iterations. This can even lead to the training corrupted. The noise accumulation problem thus must be addressed.

An intuitive solution to noise accumulation is to reduce domain gap. This can indeed reduce label noise. However, in practice, the domain gap cannot be thoroughly eliminated, so the label noise can

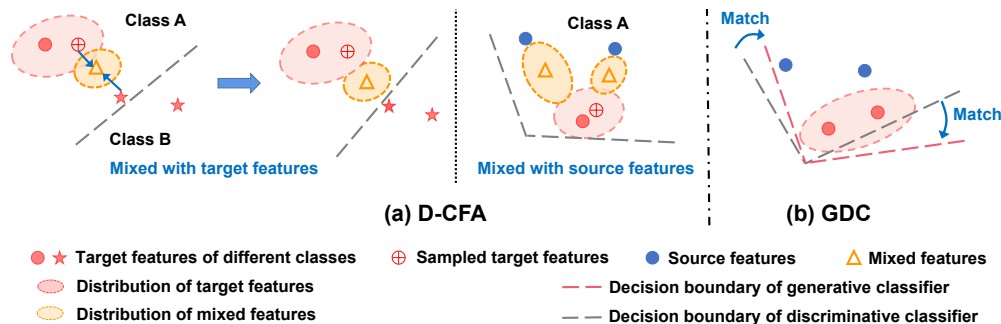

Figure 1: Illustration for our GeMo-NoRT. The GeMo-NoRT includes (a) Distribution Based Class-wise Feature Augmentation (D-CFA) and (b) Generative and Discriminative Consistency (GDC). (a) The D-CFA uses the mixed feature (the orange triangle) to replace the original misclassified feature (i.e., the red star that should be classified to class B but with a wrong pseudo-label of class A) for classifier training. The mixed feature is obtained by mixing the original feature and the feature sampled from the target distribution of class A. In this way, the mixed feature is closer to class A, thus likelier to have a *correct label* of class A. Such a mixed feature and pseudo-label A can better match each other to train a better classifier. In the next iteration, the classifier is free to assign a new pseudo-label (e.g. class B) to the original feature to stop noise accumulation. When a labeled source instance is mixed with sampled features, the mixed features serve as intermediate features between source and target domains to bridge the domain gap in each class. (b) The GDC is a consistency regularization on the target domain data by matching the predictions of the discriminative classifier to these of the generative classifier. This regularization exploits an ensemble of target knowledge from all the generative models to train a noise-robust and domain-adaptive discriminative classifier.

still exist. As such, it is necessary to train a label noise-robust model for UDA. We address the noise-robust training from the perspective of probability: to reduce the negative impact of a noisy instance, we can maximize the joint probability of its feature $f$ and pseudo-label $\hat{y}$, i.e., $p(f, \hat{y})$, which can be achieved by $\max_f p(f|\hat{y})p(\hat{y})$ (see Eq. 4). Eq. 4 essentially assumes that the pseudo-labels are intact but the features are corrupted. So we can generate the data (i.e., features) conditioned on pseudo-labels, or we can fix the pseudo label and maximize the probability of the feature $f$ given the pseudo-label $\hat{y}$, i.e. $p(f|\hat{y})$. Practically, we 'correct'/augment the feature $f$ of a noisy instance to force it to better match the pseudo-label $\hat{y}$. This differs from other noise correction methods which assume the features are intact but the labels are corrupted. To sum up, the keys to address UDA are: 1) reducing domain gap and 2) solving the probability maximization problem.

This paper proposes a generative model based noise robust training (GeMo-NoRT) method to alleviate the pseudo label noise (by solving the probability maximization problem) and meanwhile to reduce the domain gap for UDA. The key idea is to leverage a generative model, modeling the class-wise target distribution, to help train a noise-robust and domain-adaptive discriminative classifier. Specifically, GeMo-NoRT learns a generative model to enable a target distribution based class-wise feature augmentation (D-CFA) and a generative-discriminative classifier consistency (GDC), serving as feature-level and classifier-level regularization respectively.

D-CFA learns the target-domain class-wise feature distributions using generative models like normalizing flows (Durkan et al., 2019). Then, the source domain data will be augmented by the features sampled from target-domain distribution such that the domain gap can be reduced. For the pseudo-labeled target domain data, our D-CFA also provides a simple yet effective approximate solution to the probability maximization problem to alleviate noise accumulation. As shown in Figure 1a, we transform/augment the original feature $f$ by mixing it with the 'genuine' features (sampled from target-domain distribution) of the class $\hat{y}$. The augmented feature is now likelier from the class $\hat{y}$ thus increases the joint probability. Note that the distribution-sampled features can be regarded as 'genuine' features of the class $\hat{y}$ because the class-wise distribution models the whole population in a class, making the class label of sampled features highly reliable. This ensures that almost no extra label noise is introduced by the sampled features, which is critical for alleviating noise accumulation.

GDC is a consistency regularization on the target domain data by matching the predictions of the discriminative classifier to these of the generative classifier (see Figure 1b), where the generative classifier is all the off-the-shelf generative models used as a whole for class-wise probability pre-

diction. GDC aims to minimize the prediction disagreement to improve the model robustness to label noise as (Yu et al., 2019). More interestingly, our GDC is also theoretically insightful according to the famous Ben-David domain adaptation theorem (Ben-David et al., 2010). Maximizing the generative-discriminative classifier consistency naturally minimizes the hypothesis discrepancy $\mathcal{H}\Delta\mathcal{H}$, thus improving the target domain performance. Eventually, our D-CFA and GDC serving as two different regularizations improve UDA on various UDA benchmarks.

Our contributions are: (1) we as the first attempt propose to leverage normalizing flow-based generative modeling together with CNN-based discriminative modeling to improve UDA. (2) Our proposed framework (GeMo-NoRT) incorporates a D-CFA and a GDC for training a noise-robust discriminative classifier and meanwhile reducing the domain gap. (3) Our GeMo-NoRT achieves new state of the art on three popular UDA datasets, including PACS, Digit-Five, and Office-Home.

## 2 RELATED WORK

*Single-Source Domain Adaptation (SSDA)* aims to transfer knowledge from a labeled source domain to an unlabeled target domain. The challenge in this task is the domain gap caused by the distribution mismatch between the source and target domains. To reduce domain gap, most works force a feature extractor to extract domain-agnostic features for feature distribution alignment. The feature alignment can be achieved by minimizing maximum mean discrepancy (MMD) (Gretton et al., 2012; Long et al., 2015b) or optimal transport (Bhushan Damodaran et al., 2018; Balaji et al., 2019), or by confusing a domain classifier (Ganin et al., 2016; Tzeng et al., 2017). Recent works (Na et al., 2021; Wu et al., 2020) also attempt to bridge domain gap by mixing two images from different domains to generate intermediate domains. Most related to our work is a feature augmentation method, TSA (Li et al., 2021b). TSA adopts Gaussian distribution to model the domain difference in each class so that cross-domain semantic augmentation can be performed to bridge the domain gap. Instead, we use normalizing flow (NFlow) to better model the class-wise distribution itself rather than the domain difference, and further propose a consistency regularization on the generative classifier (i.e., all the NFlow models as a whole) and the discriminative one to address noise accumulation.

*Multi-Source Domain Adaptation (MSDA)* tackles a more general scenario where multiple source domains are available. This brings different kinds of source-target domain gaps, thus more challenging. Most MSDA methods still resort to feature distribution alignment by 1) using multiple domain discriminators, e.g. MDAN (Zhao et al., 2018) and DCTN (Xu et al., 2018), or 2) minimizing moment based distribution distance between different domain pairs, e.g. M³SDA-$\beta$ (Peng et al., 2019), or 3) exploiting dynamic/multiple feature extractor(s) to better extract domain-agnostic features, e.g. MDDA (Zhao et al., 2020) and DIDA-Net (Deng et al., 2022). Instead of focusing on distribution alignment, we attend to utilizing generative models to alleviate the noise accumulation of the discriminative classifier when pseudo-labels are used. Thus, we propose a Distribution Based Class-wise Feature Augmentation (D-CFA) and a Generative-Discriminative Consistency (GDC).

*Normalizing Flow (NFlow)* is a likelihood based generative model (Dinh et al., 2014; 2016; Rezende & Mohamed, 2015), similar to generative adversarial network (GAN) (Goodfellow et al., 2014) and variational auto-encoder (VAE) (Kingma & Welling, 2013). It transforms a simple distribution like standard Gaussian to match a complex one of real data by composing several invertible and differentiable mappings. With such mappings, we can evaluate the exact probability density for new data points (Kobyzev et al., 2020), which cannot be achieved by using GAN or VAE. NFlow has been applied to a variety of generation tasks, such as image generation (Kingma & Dhariwal, 2018; Ho et al., 2019; Durkan et al., 2019) and video generation (Kumar et al., 2019). In this paper, we exploit the NFlow models to model the class-wise feature distribution so that within-class feature augmentation can be performed to bridge domain gap and alleviate noise accumulation. Furthermore, we take full use of NFlow models' characteristic of the probability density estimation, which is to regard all these class-wise NFlow models as a generative classifier for class-wise probability prediction. The prediction is then used to enforce a consistency regularization on a discriminative classifier.

## 3 METHODOLOGY

In this section, we detail our GeMo-NoRT for unsupervised domain adaptation (UDA). GeMo-NoRT incorporates a target-distribution based class-wise feature augmentation (D-CFA) and a generative-discriminative consistency (GDC). The overview of our GeMo-NoRT is shown in Figure 2.

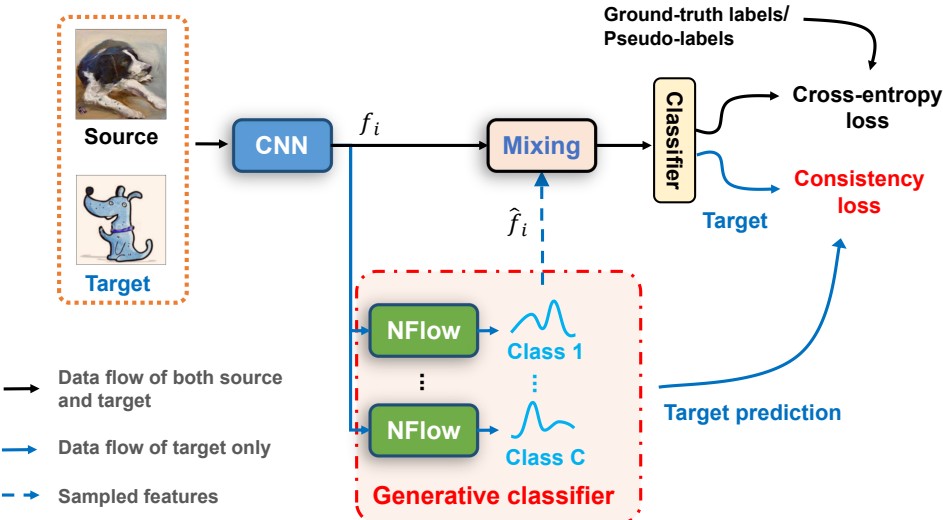

Figure 2: Overview of our GeMo-NoRT. The GeMo-NoRT first extract features $f_i$ from source and target images using a feature extractor CNN. The target features together with their pseudo-labels are then input to $C$ normalizing flows (NFlows) to model the class-wise distributions, where $C$ is the class number. These generative NFlows are used for D-CFA and GDC. The D-CFA samples a feature $\hat{f}_i$ from the $y_i$-th NFlow based distribution and mixes it with the feature $f_i$, where $y_i$ is the (pseudo-) label of $f_i$. The GDC uses all the NFlow models as a generative classifier to provide a consistency regularization for the discriminative classifier. Both the D-CFA and GDC are designed to improve the noise robustness of the discriminative classifier and meanwhile reduce domain gap. Finally, the discriminative classifier is also supervised by cross-entropy loss using the ground-truth labels of source data or the pseudo-labels of target data.

## 3.1 PRELIMINARIES

### 3.1.1 PROBLEM SETTING.

We focuses on unsupervised domain adaptation (UDA) for image classification. In UDA, we are given $K$ labeled source domains $\mathcal{S} = \{\mathcal{S}_1, ..., \mathcal{S}_K\}$ and an unlabeled target domain $\mathcal{T}$, where $K = 1$ for single-source domain adaptation (SSDA) and $K \geq 1$ for multi-source domain adaptation (MSDA). For each source domain $\mathcal{S}_k$, we have $N_{\mathcal{S}_k}$ pairs of image $x_i$ and corresponding label $y_i$, i.e. $\mathcal{S}_k = \{(x_i^{\mathcal{S}_k}, y_i^{\mathcal{S}_k})\}_{i=1}^{N_{\mathcal{S}_k}}$. For the target domain, only unlabeled images are available, $\mathcal{T} = \{x_i^{\mathcal{T}}\}_{i=1}^{N_{\mathcal{T}}}$. Following the previous MSDA works, we assume that the the same label space is shared among the source and target domains. We then aim to train a model on the training set of both source and target domains $\mathcal{S}_1 \cup ... \cup \mathcal{S}_K \cup \mathcal{T}$ so that the model can perform well on the test set of the target domain.

### 3.1.2 INSTANCE BASED FEATURE AUGMENTATION (IFA).

Before introducing our distribution based class-wise feature augmentation (D-CFA), we first revisit the naive instance based feature augmentation (IFA) as preliminaries.

IFA mixes the features of two different instances that are from the same class to bridge the domain gap. Specifically, we first extract the feature representations $\{f_i^{\mathcal{S}_k}, f_i^{\mathcal{T}}\}_{i=1}^{B}$ using a feature extractor (i.e. the backbone CNN in Figure 2), where $B$ is the batch size for each domain. Then, the augmented feature $\tilde{f}_i$ (domain-specific superscript omitted for brevity) is

$$\tilde{f}_i = \alpha f_i + (1 - \alpha) f_j,$$
$$\text{where} \quad i \neq j, y_i = y_j. \tag{1}$$

$\alpha = max(\alpha_0, 1 - \alpha_0)$, $\alpha_0$ is sampled from Beta distribution, $\alpha_0 \sim Beta(0.1, 0.1)$. If $f_i$ or $f_j$ is from the target domain, we use its pseudo-label (only high-confidence predictions are accepted as pseudo-labels in this paper while low-confidence ones are discarded).

## 3.2 DISTRIBUTION BASED CLASS-WISE FEATURE AUGMENTATION (D-CFA)

Note that the naive IFA has two limitations: 1) The features used for mixing are from the same model, thus cannot help eliminate the noise accumulation problem; 2) The features of discrete instances are not as diverse as features sampled from a distribution, limiting the diversity of augmented features. To address these two issues, we propose to leverage generative models for the distribution-based feature augmentation and introduce D-CFA.

The D-CFA mixes a feature of an instance with the feature sampled from the target distribution of the same class (modeled by the generative models). It aims to alleviate the noise accumulation and bridge the source-target domain gap in each class. This can help the classifier better adapt to the target domain.

Different from the IFA in Eq. 1, D-CFA samples the features $f_j$ from the $y_i$-th class distribution of the target domain for mixing. The class distribution is modeled by using a Multivariate Gaussian or a normalizing flow (NFlow) $\mathcal{M}_{y_i}$ (Durkan et al., 2019). Taking the NFlow for example, we train $C$ NFlow models $\{\mathcal{M}_c\}_{c=1}^C$ using the target features and pseudo-labels $\{f_i^{\mathcal{T}}, \hat{y}_i^{\mathcal{T}}\}_{i=1}^B$, with one NFlow for each class. After the NFlow models are well trained, we sample some features $\hat{f}_i$ from $\mathcal{M}_{y_i}$ for augmentation, i.e. $\hat{f}_i \sim p(f|\mathcal{M}_{y_i})$. Then, D-CFA is performed as

$$\tilde{f}_i = \alpha f_i + (1 - \alpha)\hat{f}_i. \tag{2}$$

The distribution-based continuous features $\hat{f}_i$ is more diverse than the instance-based discrete features $f_j$ in Eq. 1, so they can better bridge the domain gap in each class when mixed with source features. More importantly, $\hat{f}_i$ contains class-wise target knowledge from the generative models which can help train a noise-robust discriminative classifier (verified in Table 5). This is because compared with $f_i$, the augmented feature $\tilde{f}_i$ is closer or more similar to the 'genuine' features of the class $\hat{y}_i$, i.e. $\hat{f}_i$. In this case, $\tilde{f}_i$ is likelier to have the *correct label* of $\hat{y}_i$ than $f_i$, i.e., we augment $f_i$ to $\tilde{f}_i$ so that $\hat{y}_i$ is likelier to be a correct label of $\tilde{f}_i$. Thus, $(\tilde{f}_i, \hat{y}_i)$ brings less negative impact compared with $(f_i, \hat{y}_i)$. $\hat{f}_i$ can be regarded as a 'genuine' feature of the class $\hat{y}_i$ because $\mathcal{M}_{y_i}$ models the distribution of all the high-confidence samples (of which predicted probabilities are lager than a threshold $\tau$) in the class $y_i$. As the majority of high-confidence samples usually have clean labels, the class label of the whole population's distribution, modeled by $\mathcal{M}_{y_i}$, should be reliable.

If the label noise has been corrected, then the D-CFA plays a role as intra-class feature augmentation rather than "correcting" noisy label.

*A probability view of D-CFA against label noise.* The feature $f_i$ is usually observed with its correct label in real world. According to maximum likelihood estimation, their joint probability should be large. We thus assume that the join probability of feature $f_i$ and its pseudo-label $\hat{y}_i$ will be large if $\hat{y}_i$ is correct, otherwise $\hat{y}_i$ is noise. We then maximize their joint probability to reduce label noise:

$$\max p(f_i, \hat{y}_i) \iff \max p(\hat{y}_i)p(f_i|\hat{y}_i) \tag{3}$$

The D-CFA *fixes the pseudo-label $\hat{y}_i$ but treat $f_i$ as the random variable*, so it reformulates Eq. 3 to

$$\max_{f_i} p(f_i|\hat{y}_i). \tag{4}$$

Thus, the D-CFA augments $f_i$ to $\tilde{f}_i$ so that the probability of the feature sampled from class $\hat{y}_i$ (modeled by normalizing flow) can be maximized. In this way, D-CFA provides a simple yet effective approximate solution to the maximization problem, i.e., $\tilde{f}_i \approx \arg\max_{f_i} p(f_i|\hat{y}_i)$.

## 3.3 GENERATIVE-DISCRIMINATIVE CLASSIFIER CONSISTENCY

Recall that in D-CFA, we have the class distributions of target data modeled by $\{\mathcal{M}_c\}_{c=1}^C$. Each $\mathcal{M}_c$ can be used to predict the probability of an instance $f_i$ following the distribution of the $c$-th class, i.e. $p_c(f_i|\mathcal{M}_c)$. Therefore, we can easily obtain the normalized probability of $f_i$ on the $c$-th classes by $\frac{p_c(f_i|\mathcal{M}_c)}{\sum_{c=1}^C p_c(f_i|\mathcal{M}_c)} = \frac{\mathcal{N}(f_i|\mu_c, \Sigma_c, \mathcal{M}_c)}{\sum_{k=1}^C \mathcal{N}(f_i|\mu_k, \Sigma_k, \mathcal{M}_k)}$. Here, $\mathcal{N}(\mu_c, \Sigma_c)$ is the class prior probability parameterized by mean $\mu_c$ and covariance $\Sigma_c$. As a result, all these NFlow models $\{\mathcal{M}_c\}_{c=1}^C$ together can be regarded as a generative classifier off the shelf.

Based on this observation, we further enforce a consistency regularization on the target domain data by matching the predictions of the discriminative classifier, i.e. a fully-connected layer followed by soft-max function, to these of the generative classifier. The consistency regularization is termed as Generative-Discriminative Classifier Consistency (GDC). The GDC assumes that distilling the knowledge from the generative classifier to the discriminative one can reduce the domain gap and prevent the discriminative classifier from over-fitting the label noise (see Table 5). This assumption holds because knowledge in these two intrinsically different classifiers is usually complementary.

Denote the predictions of the discriminative and generative classifiers on the target data $\{f_i^{\mathcal{T}}\}_{i=1}^B$ as $p^D$ and $p^G$ respectively, with $p_i^G = \left\{ \frac{p_c(f_i^{\mathcal{T}}|\mathcal{M}_c)}{\sum_{c=1}^C p_c(f_i^{\mathcal{T}}|\mathcal{M}_c)} \right\}_{c=1}^C$. The GDC loss, $L_{GDC}$, is formulated as

$$L_{GDC} = \frac{1}{B} \sum_{i=1}^B ||p_i^D - p_i^G||_2^2, \tag{5}$$

where the gradient is only backward to $p_i^D$ as only the discriminative classifier is used for inference.

### 3.3.1 THEORETICAL INSIGHTS.

We further present the theoretical insights of the GDC for reducing domain gap (Ben-David et al., 2010). Given a hypothesis class $\mathcal{H}$, $\forall h \in \mathcal{H}$ we have

$$R_{\mathcal{T}}(h) \leq R_{\mathcal{S}}(h) + \frac{1}{2} d_{\mathcal{H}\Delta\mathcal{H}}(\mathcal{S}, \mathcal{T}) + \lambda, \tag{6}$$

where $R_{\mathcal{T}}(h)$ is the risk of $h$ on the target domain while $R_{\mathcal{S}}(h)$ on the source domain. And,

$$d_{\mathcal{H}\Delta\mathcal{H}}(\mathcal{S}, \mathcal{T}) = 2 \sup_{(h,h')\in\mathcal{H}^2} |\mathbf{E}_{x\sim\mathcal{S}}[\mathbb{1}(h(x) \neq h'(x))] - \mathbf{E}_{x\sim\mathcal{T}}[\mathbb{1}(h(x) \neq h'(x))]|,$$
$$\lambda = \min[R_{\mathcal{S}}(h) + R_{\mathcal{T}}(h)]. \tag{7}$$

$\mathbb{1}(\cdot)$ is a indicator function. $d_{\mathcal{H}\Delta\mathcal{H}}(\mathcal{S}, \mathcal{T})$ is the $\mathcal{H}\Delta\mathcal{H}$ distance, which plays the most important role in bounding $R_{\mathcal{T}}(h)$. The term $\mathbf{E}_{x\sim\mathcal{S}}[\mathbb{1}(h(x) \neq h'(x))]$ in the $\mathcal{H}\Delta\mathcal{H}$ distance is assumed to be a large value. This assumption holds because 1) $h$ is trained on both source and target domains, while $h'$ is trained on the target domains only; and 2) they are intrinsically different classifiers. As such, their predictions can be less consistent on source samples. Given a large $\mathbf{E}_{x\sim\mathcal{S}}[\mathbb{1}(h(x) \neq h'(x))]$, we then minimize $\mathbf{E}_{x\sim\mathcal{T}}[\mathbb{1}(h(x) \neq h'(x))]$ to approximately obtain the supreme in $d_{\mathcal{H}\Delta\mathcal{H}}(\mathcal{S}, \mathcal{T})$. In our case, $h = p^D \circ F$ and $h' = p^G \circ F(x)$, where $F$ is the features extractor. This results in

$$d_{\mathcal{H}\Delta\mathcal{H}}(\mathcal{S}, \mathcal{T}) \approx \min_{F, p^D, p^G} \mathbf{E}_{x\sim\mathcal{T}}[\mathbb{1}(p^D \circ F(x) \neq p^G \circ F(x))]. \tag{8}$$

Considering that Eq. 8 is the approximation for the $d_{\mathcal{H}\Delta\mathcal{H}}(\mathcal{S}, \mathcal{T})$, we can further minimize the approximation to reduce $\mathcal{H}\Delta\mathcal{H}$ distance. This twice minimization is equivalent to a single minimization operation in Eq. 8. In our method, the minimization problem is implemented with the GDC in Eq. 5. Thus, minimizing the GDC can reduce the $\mathcal{H}\Delta\mathcal{H}$ distance, further bound $R_{\mathcal{T}}(h)$.

### 3.4 TRAINING AND INFERENCE

#### 3.4.1 TRAINING FEATURE EXTRACTOR AND DISCRIMINATIVE CLASSIFIER.

During the training phase, the feature extractor and domain-shared discriminative classifier are supervised by cross-entropy loss as well as the GDC loss in Eq. 5.

For the labeled source data, we use the cross-entropy which is defined as

$$L_s = -\frac{1}{KB} \sum_{i=1}^{KB} \log p_{y_i^{\mathcal{S}}}(f_i^{\mathcal{S}}), \tag{9}$$

where $p_{y_i^{\mathcal{S}}}(f_i^{\mathcal{S}})$ is the $y_i^{\mathcal{S}}$-th (ground-truth) prediction of $p(f_i^{\mathcal{S}})$. $p(f_i^{\mathcal{S}})$ represents the output probability of the shared discriminative classifier for $x_i^{\mathcal{S}}$.

Table 1: SSDA results on Office-Home. The best results are in bold.

| Methods | Pr→Ar | Rw→Ar | Cl→Ar | Ar→Cl | Rw→Cl | Pr→Cl | Ar→Pr | Cl→Pr | Rw→Pr | Ar→Rw | Cl→Rw | Pr→Rw | Avg. |
|---|---|---|---|---|---|---|---|---|---|---|---|---|---|
| ResNet-50 (He et al., 2016) | 38.5 | 53.9 | 37.4 | 34.9 | 41.2 | 31.2 | 50.0 | 41.9 | 59.9 | 58.0 | 46.2 | 60.4 | 46.1 |
| DANN (Ganin et al., 2016) | 46.1 | 63.2 | 47.0 | 45.6 | 51.8 | 43.7 | 59.3 | 58.5 | 76.8 | 70.1 | 60.9 | 68.5 | 57.6 |
| CDAN (Long et al., 2017) | 55.6 | 68.4 | 54.4 | 49.0 | 55.4 | 48.3 | 69.3 | 66.0 | 80.5 | 74.5 | 68.4 | 75.9 | 63.8 |
| SymNets (Zhang et al., 2019) | 63.6 | 73.8 | 64.2 | 47.7 | 50.8 | 47.6 | 72.9 | 71.3 | 82.6 | 78.5 | 74.2 | 79.4 | 67.2 |
| RSDA-MSTN (Gu et al., 2020) | 67.9 | 75.8 | 66.4 | 53.2 | 57.8 | 53.0 | **77.7** | 74.0 | **85.4** | **81.3** | 76.5 | **82.0** | 70.9 |
| SRDC (Tang et al., 2020) | 68.7 | 76.3 | 69.5 | 52.3 | 57.1 | 53.8 | 76.3 | **76.2** | 85.0 | 81.0 | **78.0** | 81.7 | 71.3 |
| BSP+TSA (Li et al., 2021b) | 66.7 | 75.7 | 64.3 | 57.6 | 61.9 | 55.7 | 75.8 | 76.3 | 83.8 | 80.7 | 75.1 | 81.2 | 71.2 |
| GeMo-NoRT (*Ours*) | **68.8** | **80.1** | **70.3** | **59.8** | **62.6** | **57.7** | 74.8 | 75.7 | 84.3 | 79.1 | 74.7 | 77.8 | **72.1** |

For the unlabeled target data, we adopt pseudo-labels for cross-entropy calculation, following a semi-supervised learning method, FixMatch (Sohn et al., 2020). Concretely, a weakly-augmented version of a target image is input to the model to obtain its predicted class $\hat{y}_i^{\mathcal{T}}$. If the prediction is high-confidence, e.g. $q_{\hat{y}_i^{\mathcal{T}}} > \tau$ ($\tau$ is a threshold), we accept it as pseudo-label. The cross-entropy loss is then imposed on the strongly-augmented version of the same image:

$$L_u = -\frac{1}{B} \sum_{i=1}^{B} \mathbb{1}(q_{\hat{y}_i^{\mathcal{T}}} > \tau) \log p_{\hat{y}_i^{\mathcal{T}}}(f_i^{\mathcal{T}}),$$

(10)

where $q_{\hat{y}_i^{\mathcal{T}}}$ is predicted probability on the class $\hat{y}_i$. $\mathbb{1}(\cdot)$ is a indicator function.

Let us define $\lambda$ as a hyper-parameter, then the total loss is given by

$$L = L_s + L_u + \lambda L_{GDC}.$$

(11)

### 3.4.2 TRAINING NORMALIZING FLOW (NFLOW) MODELS.

The NFlow models $\{\mathcal{M}_c\}_{c=1}^{C}$ are trained to model the class distributions of the target data. They are fed with the gradient-detached features of target data $\{f_i^{\mathcal{T}}\}_{i=1}^{B}$ and their corresponding pseudo-labels $\{\hat{y}_i^{\mathcal{T}}\}_{i=1}^{B}$ – only these $\hat{y}_i^{\mathcal{T}}$ with $q_{\hat{y}_i^{\mathcal{T}}} > \tau$ have pseudo-labels, otherwise they are discarded, and then output a probability for each target instance $p(f_i^{\mathcal{T}}|\mathcal{M}_{\hat{y}_i^{\mathcal{T}}})$. Such probability is then used to maximize the log-likelihood so that the NFlow models can match the class distributions of the target domain. Equivalently, the loss $L_{NFlow}$ for optimizing the NFlow models can be defined by minimizing the negative log-likelihood, which is

$$L_{NFlow} = -\frac{1}{B} \sum_{i=1}^{B} \log(p(f_i^{\mathcal{T}}|\mathcal{M}_{\hat{y}_i^{\mathcal{T}}})),$$

(12)

### 3.4.3 INFERENCE.

We only use the discriminative classifier for inference while discard the NFlow models and D-CFA.

## 4 EXPERIMENTS

We conduct extensive experiments on a SSDA dataset, Office-Home (Venkateswara et al., 2017), and two popular MSDA datasets, including Digit-Five, PACS (Li et al., 2017), to evaluate our method. We introduce the datasets, protocols, and implementation details in the Appendix.

### 4.1 COMPARISON WITH THE STATE OF THE ART

We compare our GeMo-NoRT with following state-of-the-art methods. DANN (Ganin et al., 2016), MDAN (Zhao et al., 2018), DCTN (Xu et al., 2018), MCD (Saito et al., 2018), SymNets (Zhang et al., 2019), RSDA-MSTN (Gu et al., 2020), SRDC (Tang et al., 2020), and M³SDA-$\beta$ (Peng et al., 2019) align the distributions of each domain using domain discriminators or metric distance. CMSS (Yang et al., 2020) select source images via a curriculum manager to achieve better source-target alignment. LtC-MSDA (Wang et al., 2020) constructs a knowledge graph to capture shared class knowledge among domains for better inference. T-SVDNet (Li et al., 2021a) explores high-order correlations among multiple domains and classes by using Tensor Singular Value Decomposition (T-SVD). Such correlations are then used to better bridge domain gap. DRT (Li et al., 2021c) and DIDA-Net (Deng et al., 2022) uses a dynamic network to better achieve domain alignment.

Table 2: MSDA results on PACS. The best results are in bold. * denotes our implementation.

| Methods | ArtPaint. | Cartoon | Sketch | Photo | Avg. |
|---|---|---|---|---|---|
| Source-only | 81.22 | 78.54 | 72.54 | 95.45 | 81.94 |
| MDAN (Zhao et al., 2018) | 83.54 | 82.34 | 72.42 | 92.91 | 82.80 |
| DCTN (Xu et al., 2018) | 84.67 | 86.72 | 71.84 | 95.60 | 84.71 |
| M$^3$SDA-$\beta$ (Peng et al., 2019) | 84.20 | 85.68 | 74.62 | 94.47 | 84.74 |
| MDDA (Zhao et al., 2020) | 86.73 | 86.24 | 77.56 | 93.89 | 86.11 |
| LtC-MSDA (Wang et al., 2020) | 90.19 | 90.47 | 81.53 | 97.23 | 89.85 |
| T-SVDNet (Li et al., 2021a) | 90.43 | 90.61 | 85.49 | **98.50** | 91.25 |
| TSA* (Li et al., 2021b) | 98.25 | 81.38 | 85.95 | 88.46 | 88.51 |
| DIDA-Net (Deng et al., 2022) | 93.39 | 90.81 | 84.77 | 98.36 | 91.83 |
| GeMo-NoRT (*Ours*) | **93.55** | **91.38** | **85.72** | **98.50** | **92.29** |

Table 3: MSDA results on Digit-Five. The best results are in bold. * denotes our implementation.

| Methods | MNIST | USPS | MNIST-M | SVHN | Synthetic | Avg. |
|---|---|---|---|---|---|---|
| Source-only (Yang et al., 2020) | 92.3±0.91 | 90.7±0.54 | 63.7±0.83 | 71.5±0.75 | 83.4±0.79 | 80.3 |
| DANN (Ganin et al., 2016) | 97.9±0.83 | 93.4±0.79 | 70.8±0.94 | 68.5±0.85 | 87.3±0.68 | 83.6 |
| DCTN (Xu et al., 2018) | 96.2±0.80 | 92.8±0.30 | 70.5±1.20 | 77.6±0.40 | 86.8±0.80 | 84.8 |
| MCD (Saito et al., 2018) | 96.2±0.81 | 95.3±0.74 | 72.5±0.67 | 78.8±0.78 | 87.4±0.65 | 86.1 |
| M$^3$SDA-$\beta$ (Peng et al., 2019) | 98.4±0.68 | 96.1±0.81 | 72.8±1.13 | 81.3±0.86 | 89.6±0.56 | 87.6 |
| CMSS (Yang et al., 2020) | 99.0±0.08 | 97.7±0.13 | 75.3±0.57 | 88.4±0.54 | 93.7±0.21 | 90.8 |
| LtC-MSDA (Wang et al., 2020) | 99.0±0.40 | 98.3±0.40 | 85.6±0.80 | 83.2±0.60 | 93.0±0.50 | 91.8 |
| DRT (Li et al., 2021c) | **99.3**±0.05 | 98.4±0.12 | 81.0±0.34 | 86.7±0.38 | 93.9±0.34 | 91.9 |
| T-SVDNet (Li et al., 2021a) | **99.3**±0.11 | 98.6±0.22 | **91.2**±0.74 | 84.9±1.47 | 95.7±0.30 | 93.9 |
| TSA* (Li et al., 2021b) | 98.2±0.06 | 94.7±0.22 | 68.9±0.37 | 61.4±1.25 | 88.4±0.18 | 82.3 |
| DIDA-Net (Deng et al., 2022) | **99.3**±0.07 | 98.6±0.10 | 85.7±0.12 | 91.7±0.08 | 97.3±0.01 | 94.5 |
| GeMo-NoRT (*Ours*) | 99.2±0.05 | **98.8**±0.04 | 89.0±0.39 | **92.0**±0.54 | **97.4**±0.13 | **95.3** |

*SSDA Results on Office-Home.* In Table 1, we compare our GeMo-NoRT with other state-of-the-art methods on a single-source domain adaptation dataset, Office-Home. We can see that GeMo-NoRT beats the domain alignment based methods, e.g, SRDC (Tang et al., 2020), RSDA-MSTN (Gu et al., 2020) and SymNets (Zhang et al., 2019). Our method also surpasses the latest BSP (Chen et al., 2019)+TSA (Li et al., 2021b) by about 1% in terms of average accuracy. When the target domain is Artistic (Ar) or Clipart (Cl), GeMo-NoRT favorably outperforms all the other competitors.

Our method is less effective on Real-World (Rw) and Product (Pr) probably because the noisy pseudo-label issue on these two domains is not as severe as Ar or Cl. For instance, on Pr or Rw, the performance of these methods without noise-robust design, e.g., TSA, can easily get more than 74%, while on Ar or Cl, the accuracy are only about 60%. This implies that the noise level on Pr or Rw is much less than that on Ar or Cl. So our noise-robust design can be less effective.

*MSDA Results on PACS.* Table 2 shows that our GeMo-NoRT surpasses all the previous state-of-the-art methods on PACS, with over 1.04% performance gain. It is worth noting that LtC-MSDA (Wang et al., 2020), T-SVDNet (Li et al., 2021a), TSA (Li et al., 2021b), and DIDA-Net (Deng et al., 2022) also leverage class information by using pseudo-labels for the unlabeled target data, but both of them are inferior to our GeMo-NoRT. We owe the superiority of our GeMo-NoRT to the noise-robust training brought by the generative models' knowledge. On each specific domain, our method also achieves the best accuracy regardless of the drastic domain gap. This justifies that our GeMo-NoRT can bridge the domain gap and train a noise-robust model for MSDA.

*MSDA Results on Digit-Five.* As shown in Table 3, our GeMo-NoRT achieves the state-of-the-art performance on Digit-Five in term of the average accuracy. On the challenging USPS, SVHN, and Synthetic domains, our GeMo-NoRT manifests consistent improvements over the other methods. This shows that our GeMo-NoRT can handle the challenge of large domain gap.

## 4.2 FURTHER ANALYSIS

### 4.2.1 ABLATION STUDY ON PACS UNDER MSDA SETTING.

*Effectiveness of Instance Based Feature Augmentation (IFA).* IFA randomly mixes features of different instances in a mini-batch. To see whether it helps improve the model's performance, we compare the IFA with the baseline in the first two rows of Table 4. It is clear that the IFA increases the average accuracy of the baseline method by 1.72%. This can be explained by that randomly mixing features in a mini-batch can also reduce the domain gap.

*Significance of D-CFA.* For the D-CFA, we compare two variants, namely multivariate Gaussian based D-CFA and NFlow based D-CFA. In Table 4, we can see that Gaussian based D-CFA only achieves similar performance to the IFA. This is probably because the class-wise distribution across multiple domains can follow a much more complex distribution than Gaussian. The inaccurate Gaussian modeling may generate feature noise which hinders the effectiveness of feature mixing based augmentation. In contrast, NFlow based D-CFA is better than the former two by about 1.03%, suggesting the significance of accurate distribution modeling for feature augmen-

Table 4: Ablation study on PACS. CFA: Classwise Feature Augmentation. GDC: Generative and Discriminative Consistency. We evaluate two different distribution modelling methods: Multivariate Gaussian and NFlow.

| Methods | CFA | GDC | Avg. |
|---|---|---|---|
| Baseline | | | 88.79 |
| Instance based | ✓ | | 90.50 |
| Gaussian based | ✓ | | 90.51 |
| | ✓ | ✓ | 91.56 |
| NFlow based | ✓ | | 91.54 |
| | ✓ | ✓ | 92.29 |

tation. In addition, we show in Table 5 that the NFlow based D-CFA can reduce the noise level of pseudo-labels by integrating the knowledge in generative models into the discriminative one.

*Importance of Generative and Discriminative Consistency (GDC).* The GDC is a consistency loss that further exploits an ensemble of target knowledge from each NFlow model for training a noise-robust and domain-adaptive discriminative classifier. Table 4 shows that the GDC increases the accuracy of Gaussian based D-CFA from 90.51% to 91.56%, and NFlow based D-CFA from 91.54% to 92.29%. We owe the effectiveness of GDC to its complementary knowledge for training a noise-robust discriminative classifier and reducing domain gap.

### 4.2.2 NOISE LEVEL ANALYSIS.

We argue that our GeMo-NoRT can reduce label noise. To support this argument, we estimate the noise level of the pseudo-labels generated by the discriminative classifier in Table 5. We can observe that the baseline method accepts 94.12% pseudo-labels of the target training data but about 41.81% pseudo-labels are inaccurate. These high-confidence but inaccurate predictions demonstrates that the discriminative classifier has over-fitted the label noise. As a result, the baseline obtains a poor test accuracy of 75.02% on the Sketch domain of PACS. The D-CFA reduces the amount of high-confidence instances to avoid over-confident on some noisy instances, and also increases the accuracy of the predictions for all the target training data. This can decrease the noise level by 8.01% (from 41.81% to 33.80%), which contributes to better test accuracy. The GDC further decreases the noise level by 6.70% (from 33.80% to 26.90%). Correspondingly, the accuracy on the test set is increased considerably. These observations illustrate that with the knowledge from the generative classifiers integrated in, the discriminative classifier less over-fits the noisy labels, thus more noise-robust. This further underpins good performance on the test set of the target domain.

We further present more analysis and visualization of features in the Appendix.

## 5 CONCLUSION

This paper proposed a method named GeMo-NoRT to simultaneously alleviate the target pseudo-label noise and the domain gap for MSDA. To achieve these goals, GeMo-NoRT models the class-wise distribution of the target data using generative models to enforce a D-CFA and a GDC. Extensive experiments show that our GeMo-NoRT achieves state-of-the-art performance in all setups.

Table 5: Comparison of GeMo-NoRT (D-CFA+GDC) with the baseline on the Sketch domain of PACS. RHCP (%): the Ratio of High-Confidence Predictions (accepted as pseudo-labels) from discriminative classifier. Noise (%): the noise level in these pseudo-labels (i.e., high-confidence predictions). Acc-P (%): the accuracy of all the predictions from the discriminative classifier.

| Methods | RHCP | Noise | Acc-P | Test Acc. |
|---|---|---|---|---|
| Baseline | 94.12 | 41.81 | 57.59 | 75.02 |
| D-CFA (NFlow based) | 92.94 | 33.80 | 65.39 | 81.96 |
| GeMo-NoRT (D-CFA+GDC) | 85.79 | 26.90 | 69.19 | 85.72 |

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

## A  APPENDIX

The Appendix is organized as follows. Section A.1 revisit the Transferable Semantic Augmentation (TSA) (Li et al., 2021b). Section A.2 provides experimental settings, including datasets. protocols, and implementation details. Section A.3 presents further analysis for our GeMo-NoRT. Section A.4 shows the t-SNE visualization (Maaten & Hinton, 2008) of the learned features.

### A.1  REVISIT TSA

Transferable Semantic Augmentation (TSA) (Li et al., 2021b) adopts Gaussian distribution to model the domain difference in each class so that cross-domain semantic augmentation can be performed to bridge the domain gap.

Denote $\mu_c^{\mathcal{T}}, \mu_c^{\mathcal{S}}$ as the mean of target and source domain features in the class $c$, respectively. Here, pseudo-label for the target domain is obtained as $\hat{y} = \arg\max p^{\mathcal{T}}$. TSA then model the inter-domain difference, i.e., $\Delta\mu_c = \mu_c^{\mathcal{T}} - \mu_c^{\mathcal{S}}$, using Gaussian distribution $\mathcal{N}(\Delta\mu_c, \Sigma_c^{\mathcal{T}})$. $\Sigma_c^{\mathcal{T}}$ is the intra-class covariance of the target domain. After that, TSA augment source feature $f_i^{\mathcal{S}}$ by

$$\tilde{f}_i^{\mathcal{S}} \sim \mathcal{N}(f_i^{\mathcal{S}} + \lambda\Delta\mu_c, \lambda\Sigma_c^{\mathcal{T}}), \tag{13}$$

where $\lambda$ is a hyper-parameter.

TSA only focus on reducing domain gap by applying augmentation to *source domain features*. In contrast, our GeMo-NoRT can not only reduce domain gap, but also deal with target label noise by applying augmentation to *target features*. In addition, TSA model the domain difference using Gaussian distribution. Instead, we use normalizing flow (NFlow) to better model the class-wise distribution itself rather than the domain difference, and further propose a consistency regularization on the generative classifier (i.e., all the NFlow models as a whole) and the discriminative one to address noise accumulation.

## A.2 EXPERIMENTAL SETTING

*Datasets and Protocols.* **Office-Home** (Venkateswara et al., 2017) includes four domains, namely Clip Art (Cl), Artistic (Ar), Real-World (Rw), and Product (Pr). It has 15,500 images of 65 categories. We follow the single-source setting in (Tang et al., 2020) to evaluate our method on all the 12 transfer tasks. **Digit-Five** is a MSDA datset which comprises five domains, namely USPS, MNIST (Lecun & Bottou, 1998), MNIST-M (Ganin & Lempitsky, 2015), Synthetic Digits (Ganin & Lempitsky, 2015), and SVHN (Netzer et al., 2011). The protocol is the same as (Peng et al., 2019): All the 9,298 images of USPS are used for training when it is used as a source domain; For the rest four domains, 25,000 images are randomly sampled for training while 9,000 images for testing. **PACS** consists of four domains, including Cartoon, Photo, Sketch and Art Painting, totally 9,991 images of 7 categories. The official train-val splits in (Li et al., 2017) are adopted for MSDA task. For MSDA, we take turns choosing one domain as target and the rest as source domains.

We train the model on the training set of source and target domains, and report the average results of three runs on the test set of the target domain.

*Implementation Details.* On Office-Home, we adopt ImageNet pre-trained ResNet-50 as backbone, following (Tang et al., 2020). We then optimize the model for 100 epochs with SGD. We set the initial learning rate to 0.001, batch size to 32 for each domain, and the threshold $\tau$ in Eq. (11) is 0.95. On Digit-Five, we use the same backbone as (Peng et al., 2019): a CNN with three convolution layers followed by two fully connected layers. We train the model with SGD for 30 epochs. We set the initial learning rate to 0.05 and decay it using the cosine annealing strategy (Loshchilov & Hutter, 2016). The batch size $B$ is 64 for each domain, and the threshold $\tau$ is 0.75. For the backbone model used on Digit-Five and Office-Home, we further incorporate a fully-connected layer as a bottleneck to reduce the feature dimension from 2048 to 512. This can reduce the computational cost for training the NFlow models. On PACS, we follow (Wang et al., 2020) and adopt an ImageNet pre-trained ResNet-18 (He et al., 2016) as backbone. We optimize the model with Adam (Kingma & Ba, 2014) for 100 epochs. We set the initial learning rate 5e-4, batch size $B = 16$, and $\tau = 0.95$.

For all the experiments, we adopt neural spline flow (Durkan et al., 2019) as our NFlow model. The NFlow model stacks three blocks, with each block comprising an actnorm, invertible 1x1 convolution (Kingma & Dhariwal, 2018) and a four-layer multi-layer perceptron (MLP) as coupling transform. We use Adam optimizer with an initial learning rate of 5e-4 to train the NFlow. We exploit a memory module to cache the latest features to facilitate end-to-end training since there can be no instance of some classes in a mini-batch. We apply the D-CFA and GDC after 10 epochs so that the NFlow models are properly trained. The weight for GDC loss, i.e. $\lambda$ in Eq. (12), is 0.5.

All our experiments are based on PyTorch (Paszke et al., 2017; 2019) and Dassl codebase (Paszke et al., 2017; Zhou et al., 2020) (`https://github.com/KaiyangZhou/Dassl.pytorch`). Our code will be released.

## A.3 FURTHER ANALYSIS FOR GEMO-NORT

In this section, we will provide more experimental results and further analysis for GeMo-NoRT on PACS under MSDA setting.

### A.3.1 FURTHER ANALYSIS ON GDC.

*Superiority of Using Generative Classifier in GDC.* The GDC in Eq. 5 exploits the generative classifier to apply consistency regularization on the discriminative one. We also compare this design choice with using the discriminative classifier itself for the consistency, which is termed self-consistency. Concretely, we adopt label smoothing strategy to the predictions of the discriminative

Table 6: The GDC vs. self-consistency (using the discriminative classifier itself for consistency regularization).

| Methods | Avg. |
|---|---|
| GDC (generative classifier) | 92.29 |
| Self-consistency | 90.65 |

Table 7: Apply the GDC on all the target data vs. on only the disagreed target data between generative and discriminative classifiers.

| Methods | Avg. |
|---|---|
| GDC on all the target data | 92.29 |
| GDC on disagreed target data | 91.86 |

Table 8: Evaluation on different metric functions in GDC. Results reported on PACS under MSDA setting.

| Methods | Avg. |
|---|---|
| L2 distance | 92.29 |
| L1 distance | 90.35 |
| KL divergence | 89.50 |

Table 9: Modelling only target distribution vs. modelling all domains'. Results reported on PACS under MSDA setting.

| Methods | Avg. |
|---|---|
| GeMo-NoRT (target domain) | 92.29 |
| GeMo-NoRT (all domains) | 90.55 |

classifier and use the smoothed predictions to enforce consistency on non-smoothed predictions. Table 6 shows that the GDC outperforms the self-consistency, due to its complementary knowledge used for alleviating noise accumulation and reducing domain gap.

*Whether All the Target Data Needed for Applying GDC?* In practice, we apply the GDC to all the target data. Here we further evaluate applying the GDC only on these target data of which the predictions are disagreed by generative and discriminative classifiers. As shown in Table 7, we can see that only using disagreed data (i.e., without using agreed data) decreases the accuracy on PACS by 0.43% but is still better than without GDC, i.e., 91.54% in Table 4. These two comparisons demonstrate that both the agreed data and disagreed data contribute to the better performance of GDC. The GDC on the agreed data can possibly prevent the discriminative classifier from over-fitting label noise, while the disagreed data can help reduce the domain gap as per Eq. 8.

*Alternative metric function for $L_d$.* The GDC adopts L2 distance by default. We also investigate L1 distance and KL divergence in Table 8. It is clear that GDC based on L2 distance works better than L1 distance with a 1.94% gap, and beats the version based on KL divergence by 2.79%.

### A.3.2 MODELING ONLY TARGET DISTRIBUTION VS. MODELING ALL DOMAINS' DISTRIBUTIONS.

By default, we only model the target distribution of each class using NFlow. We further compare this design choice with modeling all the domains' distribution of each class. The comparative results on PACS are shown in Table 9. We find that only modeling the target distribution is clearly better than modeling all, possibly because the generative models trained on only the target domain are less source-biased (i.e., not dominated by source data). This can make these generative models more complementary to the source-biased discriminative classifier, further contributing to better performance.

### A.3.3 SENSITIVITY OF HYPER-PARAMETER.

In Eq. (12), there is a hyper-parameter $\lambda$ in our GeMo-NoRT which is the loss weight for the GDC. We evaluate the sensitivity of the performance with respect to $\lambda$ in Figure 3. We can see that the performance is generally insensitive to $\lambda$, with $\lambda = 0.5$ leading to the best performance of 92.29%. The performance of $\lambda \in [0.3, 0.7]$ is consistently better than without GDC (i.e., $\lambda = 0$), which manifests the effectiveness of the GDC.

### A.4 VISUALIZATION.

We further provide the visualization of final features in Figure 4 to better understand the D-CFA and GDC in our GeMo-NoRT. We can observe that incorporating the D-CFA to the baseline model leads to less features lying around the decision boundary (see the top left area of Figure 4a and Figure 4b). The GeMo-NoRT further includes the GDC to prevent the discriminative classifier from

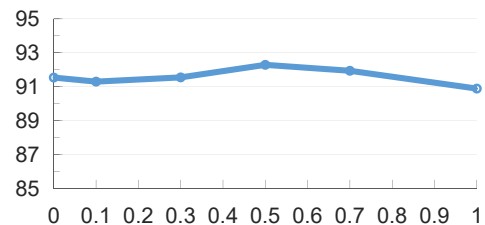

Figure 3: Sensitivity of $\lambda$ for GDC on PACS dataset.

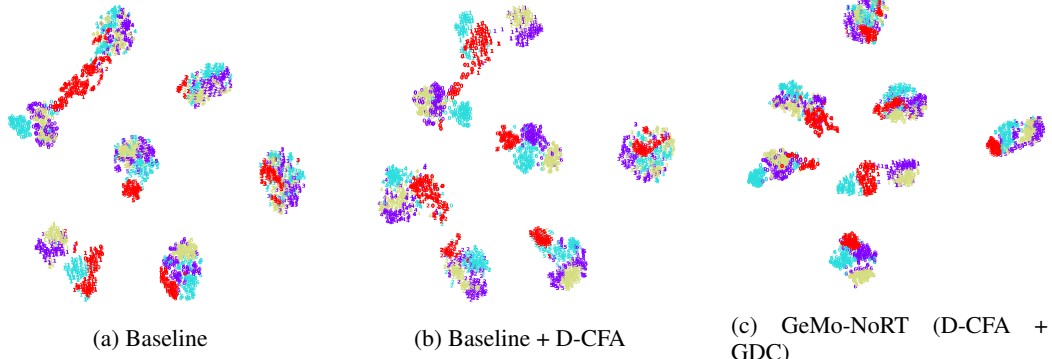

(a) Baseline        (b) Baseline + D-CFA        (c) GeMo-NoRT (D-CFA + GDC)

Figure 4: Visualization of features from baseline (FixMatch), baseline+D-CFA and GeMo-NoRT on PACS using t-SNE (Maaten & Hinton, 2008). Red color denotes the target domain of Sketch while other colors denote source domains. Different digits (0-6) represents different classes. Best viewed with zoom-in.

over-fitting the label noise. As depicted in Figure 4c, incorporating GDC to our method can increase the separability of different classes and better align different domains. As a result, our GeMo-NoRT achieves the best classification performance.

