# OpenReview forum: "Generative Model Based Noise Robust Training for Unsupervised Domain Adaptation"
_ICLR.cc/2023/Conference — Submitted to ICLR 2023_

### Official Review · Reviewer_YwCr · 2022-10-24

**Confidence:** 5
**Correctness:** 2
**Technical Novelty And Significance:** 2
**Empirical Novelty And Significance:** 2
**Recommendation:** 3

**Clarity, Quality, Novelty And Reproducibility:**

As discussed above, insufficient novelty is contained in the proposed method. Clarity and reproducibility lack sometimes, among others for the above weaknesses.

**Strength And Weaknesses:**

Strength:
1. The paper is nicely written and the proposed method is well described.
2. Useful illustrations are provided that illustrate essential parts of the problem or method.
3. The empirical evaluation is thorough and conducted in diverse settings, including single- and multi-source settings.

Weaknesses:
1. The main concern for me is the difference with TSA (Li et al. 2021b). First, the idea of using the class-wise target distribution to perform augment is similar to TSA. There should include more comparisons and discussions. Second, this work exploits NFlow instead of Gaussian distribution to model the class-wise distribution of target data but the motivation is not clear and is not well-proven. For example, in Relate Work section, the authors claim that NFlow models can be performed to bridge the domain gap and alleviate noise accumulation. How about Gaussian-based models?  Third, it would be nice to revisit TSA before introducing the proposed D-CFA.

2. Concern about noise correction. From Fig. 1(a), we can see that target samples around the decision boundary of the discriminative classifier (red star) are prone to be mixed. My concern is that would these samples be corrected over and over again.

3. More explorations are needed. It is a bit unclear that two assumptions "the pseudo-labels are intact but the features are corrupted" and "the features are intact but the labels are corrupted" are in the Introduction.

4. A doubt about the feasibility of class-wise distribution. D-CFA samples the features from the class distribution of the same class (depending on pseudo labels predicted by the generative models), however, due to domain shift, untrustworthy pseudo-labels may affect the construction of the distribution. Thus, the sampled features might be incorrect.

5. About experimental results. First, the improvement on Office-Home is limited (72.1 vs TSA: 71.2 & SRDC: 71.3) and more recent results should be included. Second, it might be better to provide the results of TSA on MSDA in Table 3 and Table4.

**Summary Of The Paper:**

This work proposes a new domain adaptation method for both single-source and multi-source conditions. The main idea is to alleviate the error accumulation of target pseudo labels and reduce the domain gap at the prediction level. Technically, they use a generative model (normalizing flows) to model the class-wise distribution of the target data and solve a probability maximization problem. Experiments on three benchmarks demonstrate the effectiveness.

**Summary Of The Review:**

In summary, the paper proposes an interesting approach to a relevant problem. Clarity and reproducibility could be improved with a minor revision. At this point, my main concern is the difference with TSA. This point requires clarification and some solid empirical support before warranting acceptance of this paper.

---

> ### Author Response · Authors · 2022-11-18
> **Response to Reviewer YwCr**
>
> Q1: Compare with TSA.
>
> A: We have compared our method with TSA in Sec. 2 (highlighted in blue) and Tab. 1. We have also include results of TSA on MSDA in Tab. 2, 3, where our method  surpasses TSA by 6.78\% on PACS (92.29\% vs. 88.51\%) and 13\% on Digit-Five (95.3\% vs. 82.3\%).
> We highlight that TSA use Gaussian distribution to model the domain difference in each class while we use NFlow to better model the class-wise distribution itself; TSA aims at reducing domain gap while our D-CFA focuses more on alleviating label noise. We further propose a consistency regularization on the generative classifier and the discriminative one to address noise accumulation.
>
> Q2: The motivation of using NFlow rather than Gaussian.
>
> A: In Sec. 3.2 and Tab. 4, we show that Gaussian distribution can be our choice, but it worked worse than NFlow. We thus analyze in Sec. 4.2.1 that the worse performance of Gaussian is probably "because the class-wise distribution across multiple domains can follow a much more complex distribution than Gaussian" --- the class distribution among a single domain can be Gaussian, but when across multiple domains, this assumption may be less trustworthy due to domain gap.
>
> Q3: Revisit TSA before introducing D-CFA.
>
> A: We have included it in Sec. A.1. Besides, as the answer to Q1, we have discussed and compared with TSA in the main paper.
>
> Q4: Concerns about noise correction.
>
> A: These samples can be "corrected" only if they are noise. If they are not, i.e., these two samples for mixing are from the same class, D-CFA will play a role as intra-class feature augmentation rather than "correcting" these samples' labels. We have included this in Sec. 3.2.
>
> Q5: The assumption regarding the pseudo-labels and the features.
>
> A: Follow this sentence, we stated that "we can generate the data (i.e., features) conditioned on pseudo-labels". This means that we fully trust the pseudo-labels as correct ones but assume that the noisy issue is caused by the corrupted features. This is why we try to generate new features from NFlow to replace the "corrupted" ones given the "correct" pseudo-labels.
>
> Q6: Untrustworthy pseudo-labels may affect the construction of the distribution. Sampled feature thus may be incorrect.
>
> A: Please see Q4 of Reviewer j3cT.
>
> Q7: Limited improvement on Office-Home.
>
> A: Please see Q2 of Reviewer 8FeU.

---

### Official Review · Reviewer_j3cT · 2022-10-24

**Confidence:** 4
**Correctness:** 2
**Technical Novelty And Significance:** 2
**Empirical Novelty And Significance:** 2
**Recommendation:** 5

**Clarity, Quality, Novelty And Reproducibility:**

<Clarity>

- This manuscript is well-organized and easy to follow.


<Quality>

- It is not clear why utilizing NFlow models can reduce label noise in pseudo-labeled data. Since NFlow models are trained based on pseudo labels, the model may learn inaccurate class-wise data distribution. It should lead to degrade the performance of the discriminative classifier, which will make label noise in pseudo-labeled data much worse.
- In Section 3.2, the authors state "We further assume that y_i and \hat{y}_i are independent," but is it reasonable? In many cases, x_i stems from p(x|y_i), and \hat{y}_i is determined based on x_i. Consequently, there should be some depedency between these two.
- Class prior probabilities are not considered in the computation of p^G. Why is it?
- Just after Eq. (9), the author state "we assume that their discrepancy should be intrinsically large to fit the maxpD,pG part," but this is not reasonable, because neither classifier is not trained to maximize the discrepancy between pG and pD. On the contrary, the discriminative classifier is trained to make pD consistent to pG, which is apparently doing the opposite thing.
- From Table 5, it is not clear to see if the proposed algorithm really improves the accuracy of pseudo labels, because the threshold \tau seems to be fixed. In addition, recent methods such as [R1] and [R2] adopt some sophisticated strategies to determine the threshold for more accurate pseudo labels. It would be better to compare the proposed method with such methods to show the effectiveness of D-CFA+GDC.

[R1] "AdaMatch: A Unified Approach to Semi-Supervised Learning and Domain Adaptation," ICLR 2022.

[R2] "Dash: Semi-Supervised Learning with Dynamic Thresholding," ICML 2021.


<Novelty>

Somewhat limited. The proposed method is based on FixMatch (as well as ICT[R3]), and one of the sample pair to mixup is replaced from a real sample to a generated one from NFlow. Using NFlow models might provide some novelty, but there seems to be no ingenuity in how to effectively utilize NFlow models specifically for domain adaptation.

[R3] "Interpolation Consistency Training for Semi-Supervised Learning," IJCAI 2019.


<Reproducibility>

Good. Several important implementation details are provided in the appendix, and the authors state "Our code will be released."


**Strength And Weaknesses:**

<Strength>

- The proposed method seems to be applicable to a wide range of model architectures.
- This manuscript is well-organized and easy to follow.


<Weakness>

- The proposed method requires training of several NFlow models, which may lead to prohibitive computation cost especially when the number of classes is large. In spite of this, the performance gain seems not so significant in the experiments.
- To train the NFlow models, we also need a sufficiently large number of target samples for every class, which may limit the applicability of the proposed method in practice.


**Summary Of The Paper:**

This paper presents a novel UDA method that utilizes generative models trained in the feature space to alleviate a domain gap. The generative model is trained in a feature space for each subset of the target data categorized into a common pseudo-class, and generated samples from this model are used for mixup-like augmentation of extracted features. Learning with this augmentation effectively reduces a domain gap, which makes a classifier trained on top of the feature perform well on the target data. Experimental results with several popular benchmark datasets show that the proposed method performs on par or better than state-of-the-art UDA methods.


**Summary Of The Review:**

The design of the proposed algorithm is interesting, but I have several concerns on its validity. I vote for "weak reject."

---

> ### Author Response · Authors · 2022-11-18
> **Response to Reviewer j3cT**
>
> Q1: Computational cost.
>
> A: The computation cost is not much as we only applies NFlow models to feature space, which is low dimensional compared with image space. So we can train the model on the 65 classes of Office-Home. If the number of classes further increases, we can simply force all the NFlow models share some bottom layers or use Gaussian as in Tab. 4 to save computational cost.
>
> Q2: The performance gain is not significant.
>
> A: Please see Q2 of Reviewer 8FeU.
>
> Q3: Large number of target samples for every class are required.
>
> A: If there are limited number of target samples, we can reduce the layers (parameters) of NFlow or simply use a Gaussian as the generative model to implement our method. A simple Gaussian usually requires much less target samples for training. In Tab. 4, we show that using Gaussian can also improve the performance for UDA considerably.  Moreover, in UDA, the target samples are unlabeled, so it probably safe to assume that we have sufficient target samples for training.
>
> Q4: Why normalizing flow can reduce label noise?
>
> A: The NFlow can be less noise-biased in two-fold. First, the NFlows are trained on the high-confidence predictions with probability > $\tau$ (only in this case we accept them as pseudo-labels)  (see Sec. 3.2). This can largely reduce label noise as low-confidence samples are discarded. Furthermore, NFlows model the distribution of the whole population for each class, so they can less biased to a small portion of label noise in these high-confidence pseudo-labels.
>
> Q5: The assumption on the independence of $y$ and $\hat{y}$.
>
> A: Thanks. Please refer to Q1 of Reviewer ahQT. We have revised the paper to avoid such a strong assumption.
>
> Q6: Class prior probability not used to compute $p^g$.
>
> A: Sorry for this confusion. We do use class prior probabilities to compute $p^G$ as the exact likelihood estimation in each NFlow needs the class prior. We updated Sec. 3.3 for this.
>
> Q7: The assumption on $\max_{p_D, p_G}$.
>
> A: Thanks. We have revised this part in Sec. 3.3.1 (highlighted in red). That is, the term $\mathbf{E}_{x\sim \mathcal{S}}[\mathbf{1}(h(x) \neq h'(x))]$ in the $\mathcal{H}\Delta\mathcal{H}$ distance is assumed to be a large value. ... we then minimize the discrepancy of $p_D, p_G$ on the target data to approximately obtain the supreme in $\mathcal{H}\Delta\mathcal{H}$ distance.
>
> Q8: The threshold issue and compare with AdaMatch\&Dash.
>
> A: (1) Our method does improve the accuracy of pseudo labels, as verified in Tab. 5. The baseline FixMatch also adopts a threshold, but the noise level in its pseudo-labels is 41.81. In contrast, our GeMo-NoRT reduce the noise level to 26.90 --- lower noise level means higher accuracy of pseudo-labels.
> (2) The motivation and the method of our GeMo-NoRT differ from both. Our method adopts generative models to deal with the label noise and domain gap, but they focus on how to adaptively set threshold. Integrating AdaMatch and Dash to our method can possibly further improve the performance. This will be our future work.
>
> Q9: No ingenuity in how to effectively utilize NFlow models specifically for domain adaptation.
>
> A: We show in the Fig. 1(a) that the D-CFA can reduce domain gap when the features sampled from NFlow models are mixed with the source domain features. And in Sec. 3.3.1, we further provide the theoretical insights on how GDC can reduce domain gap.

---

### Official Review · Reviewer_ahQT · 2022-11-01

**Confidence:** 3
**Clarity, Quality, Novelty And Reproducibility:** Details are mentioned in the Strength…
**Correctness:** 2
**Technical Novelty And Significance:** 2
**Empirical Novelty And Significance:** 3
**Recommendation:** 5

**Strength And Weaknesses:**

The authors study an important problem and propose novel techniques to deal with the noisy pseudo-label issue in UDA settings. The experiments are comprehensive and ablation studies are provided. The paper is generally well-written. However, I still have several concerns.

For the method part, I have several questions about the theoretical analysis. As a result, I doubt the effectiveness of the methods from a theoretical perspective.
1. For the probability view of D-CFA against label noise, the approximation steps are unclear and need further theoretical analysis. Firstly, the authors suppose that $y_i$ and $\hat{y}_i$ are independent, which may not be correct. The reason is that the pseudo-labels are generated to be close to the true label, making the two labels dependent. Secondly, even if the labels are independent, it is still unclear how to get Equation (4) from Equation (3). For example, why $p(f_i|\hat{y}_i,y_i)$ can be approximated as $p(f_i|\hat{y}_i)$?
2. For the theoretical insights in Section 3.1.1. I do not agree with the authors' claim on the $H\Delta H$ divergence. Since $h$ and $h'$ are optimized according to the sup function in the divergence, they are not trained on the same labeled source function. As a result, it may not be correct to claim that the distance can be approximated by the supreme in the target distribution.

For the experiment part.
1. I do not understand the results in the Office-Home dataset. The proposed method achieves superior results when the target domain is Ar or Cl while they are not the highest results when the target domain is Pr or Rw. Could the authors explain this phenomenon?

Minor issues:
1. The third line in Section 3.1.1: $k1$ $\rightarrow$ $K > 1$

**Summary Of The Paper:**

The authors propose a novel method to handle the noisy pseudo-label issue in UDA settings. They first train normalizing flow models to estimate the distribution of the target domain bases on generated pseudo-labels. Then they use the generative models to construct the D-CFA on features in the source domain. Besides, they use the GDC loss to require that predictions of the discriminative model and generative model (i.e., the normalizing flows) are consistent. Finally, they conduct experiments on several datasets to prove the effectiveness of the methods.

**Summary Of The Review:**

As mentioned in the Strength and Weakness section, I think the experiments are comprehensive while the theoretical analysis needs further explanation. As a result, I am negative about the paper in this round.

---

> ### Author Response · Authors · 2022-11-18
> **Response to Reviewer ahQT**
>
> Q1: Concerns regarding the probability view of D-CFA against label noise.
>
> A: Indeed, this is a strong assumption, which may not true in practice. We have revised this part to make it more reasonable. We change it to maximizing the joint probability of feature and its pseudo-label $p(f_i, \hat{y}_i)$ to reduce label noise. This is equivalent to $\max p(\hat{y}_i)p(f_i|\hat{y}_i)$. D-CFA then fixes the pseudo-label $\hat{y}_i$ but update $f_i$ to approximately maximize $p(f_i|\hat{y}_i)$.
>
> Q2: The GDC and $\mathcal{H}\Delta\mathcal{H}$.
>
> A: Thanks for the advice. We have corrected and highlighted it in red in Sec. 3.3.1: "The term $\mathbf{E}_{x\sim \mathcal{S}}[\mathbf{1}(h(x) \neq h'(x))]$ in the $\mathcal{H}\Delta\mathcal{H}$ distance is assumed to be a large value. This assumption holds because 1) $h$ is trained on both source and target domains, while  $h'$ is trained on the target domains only; and 2) they are intrinsically different classifiers. As such, their predictions can be less consistent on source samples."
>
> Q3: Results on Pr or Rw domains of Office-Home.
>
> A: Our method is less effective on Pr or Rw probably because the noisy pseudo-label issue on these two domains is not as severe as Ar or Cl.  For example, on Pr or Rw, the performance of these methods without noise-robust design, like TSA, can easily get more than 74\% while on Ar or Cl, the accuracy are only about 60\%. This implies that the noise level on Pr or Rw is much less than that on Ar or Cl. So our noise-robust design can be less effective.

---

> > ### Comment · Reviewer_ahQT · 2022-12-04
> > **Response to Authors**
> >
> > I thank the authors' response while my main concerns remain. The theoretical analysis has several approximation steps and more formal results should be provided. As a result, I will keep my score unchanged.

---

> > > ### Author Response · Authors · 2022-12-07
> > > **Response to Reviewer ahQT**
> > >
> > > A: Thanks for the comment. The approximation steps are generally reasonable and common in theoretical analysis. Even if there are approximation steps, our method can reduce the $\mathcal{H}\Delta \mathcal{H}$ to alleviate the domain gap for MSDA task and improve the performance. The experimental results are also included in the section of Experiments which verify the effectiveness of our method.

---

### Official Review · Reviewer_8FeU · 2022-11-06

**Confidence:** 4
**Correctness:** 3
**Technical Novelty And Significance:** 2
**Empirical Novelty And Significance:** 2
**Recommendation:** 5

**Clarity, Quality, Novelty And Reproducibility:**

The work proposed a class-wise NFlow-based UDA method. They introduced a new generative model based  "Feature Mix-Up" scheme (D-CFA), and a consistency constraint (GDC) on these NFlow-based and the task classifier. The work has enough novelty.


In the paper, they theoretically and experimentally justified that the proposed techniques could decrease the noise level of pseudo labels ( 8.01% by D-CFA, and 6.70% by GDC)

**Details Of Ethics Concerns:**

Nil

**Strength And Weaknesses:**

Strength:

- The work proposed a generative model based $\\textit{noise-robust}$ UDA method. They introduced an NFlow-based feature Mix-Up scheme (D-CFA): Instead of using instance features, they use the classwise NFlow models to sample features to do feature mix-up. It can provide enough diversity while eliminating the noise from instance features. Also, they proposed a consistency constraint (GDC) on these NFlow-based classifiers and the task classifier.
- The theoretical justification for GDC and theoretical discussions from the perspective of probability about D-CFA are given in the paper.
- Their experiment results showed that GDC decreased the noise level by 8.01% and D-CFA further decreased the noise level by 6.70%.

Weakness:

-  I have some concerns about the theoretical discussions of  GDC in the paper. The definition of  $=>$ in EQ3 is unclear. If it is $\geq$, the deduction has to be given.
- The experiment results are not very strong. Though the proposed method can largely decrease the noise level, the final classification performance was only marginally improved by 1% compared to SOTA.



**Summary Of The Paper:**

This work proposed an NFlow-based noise-robust UDA method. Specifically, they trained an NFlow (Durkan et al. 2019)  generative model of the feature distribution for each class by using the noisy pseudo labels generated from the source. With the classwise NFlow models, they introduced a modified feature mix-up scheme (i.e. D-CFA), and a consistency regularization loss on these NFlow-based classifiers and the task classifier (i.e. GDC). The experiments show that the proposed method can improve the DA performance by 1% compared to the SOTA methods on office-home for single-source UDA, as well as MSDA and PAC for multi-source UDA.

**Summary Of The Review:**

In summary, the work proposed a novel generative model based noise-robust UDA method. They theoretically and experimentally justified that the proposed techniques can decrease the noise level of pseudo labels. However, there are still some concerns about the method and experiment results.

---

> ### Author Response · Authors · 2022-11-17
> **Response to Reviewer 8FeU**
>
> Q1: $\Rightarrow$ in Eq. (3).
>
> A: Sorry for the confusion. It means "equivalent to". We have updated Eq. (3) now.
>
> Q2: Marginal improvement over SOTA.
>
> A: Our method achieves about 1\% improvement over the previous SoTA on Office-Home and Digit-Five. In contrast, TSA only beats its SoTA competitor GVB-GD by 0.8\% on Office-Home and DIDA-Net is better than T-SVDNet by 0.6\% on Digit5. Their improvement even less significant than ours.
> We have verified in ablation study that our method can improve the baseline FixMatch by 3.5\%. This clear improvement can justify the effectiveness of our method.
> Importantly, we contribute to the community by providing a novel generative model-based augmentation method to simultaneously train noise-robust model and reduce the domain gap for MSDA.

---

> > ### Comment · Reviewer_8FeU · 2022-11-24
> > **Concerns regarding the probability view of D-CFA against label noise**
> >
> > The  theoretical justification for  D-CFA  is not convicing. There is no guarantee that $p(f_i, y_i) $  can be improved by maximizing $p(f_i, \widehat{y_i})$ in EQ3&4. It cannot be assummed that $p(f_i, \widehat{y_i})$ will be large if $\widehat{y_i}$ is correct.  The method highly depends on how good the pesudo labels are and how robust the NFlow models can be trained.  Actually, pesudo labels generated by source models can be heavily noisy due to domain shifts. A large proportion of pesudo labels can be misleading, some even with high confidence.

---

> > > ### Author Response · Authors · 2022-11-29
> > > **Response to Reviewer 8FeU**
> > >
> > > Q1: It cannot be assumed that $p(f_i,\hat{y}_i)$ will be large if $\hat{y}$ is correct.
> > >
> > > A: This assumption holds because, in the real world, the feature $f_i$ is usually observed with its correct label. Therefore, according to maximum likelihood estimation, the joint probability of feature $f_i$ and its pseudo-label $\hat{y}_i$, i.e., $p(f_i,\hat{y}_i)$, is large if $\hat{y}_i$ is correct. Here, the pseudo-label $\hat{y}$ in Eq. (3) is a **random variable** which can be correct (ground truth) or wrong label.
> > >
> > >
> > > Q2: Pseudo-labels used for NFlow training can be noisy due to domain shift.
> > >
> > > A: (1) Even if pseudo-labels contain noise, the class label of an NFlow is usually more robust than the pseudo-label of a single sample because it models the distribution of all the samples in a class. This means that the NFlow's class label is likelier to be correct as long as the majority of pseudo-labels of a class are correct. We can observe in Tab. 5 that most pseudo-labels are correct (ACC-P>57). So, it is usually safe to assume that the whole population of a class used for NFlow training is more reliable and robust than the pseudo-label of a single sample. (2) Using NFlow to model the target distribution can also help **reduce domain shift when mixed with source features**. As the training goes on, the domain shift will be gradually reduced by D-CFA, leading to better pseudo-labels for noise-robust training. As such, the D-CFA can reduce domain shift by mixing with source features and meanwhile train a noise-robust model by mixing with target features. These two goals can benefit from each other. (3) We do not aim to eliminate label noise completely. Instead, we seek to train a label noise-robust model with better performance when the noise exists. Our experimental results in Tab. 4&5 have verified that our method can reduce label noise and achieve better performance.

---

### Decision · Program_Chairs · 2023-01-20

**Decision:**

Reject

**Justification For Why Not Higher Score:**

N/A

**Justification For Why Not Lower Score:**

N/A

**Metareview: Summary, Strengths And Weaknesses:**

Strength aspects of the proposed approach relate to a certain novelty (not unanimously) and as containing some theoretical contributions, also corroborated by a complete experimental stage. The paper was also considered in general well presented, but still subject to improvement.

Conversely, weak points regard some parts to be clarified concerning the technical and theoretical contributions, the latter containing hypotheses or approximations not better motivated or explained, justifications of the approach not so well highlighted, and just slightly increased performance wrt the state of the art, especially when compared to the heavier computational cost. Indeed, novelty was argued by a couple of reviewers.

Rebuttal was provided, but reviewers were not fully convinced of some of the replies, and none changed his/her ratings, which resulted all below threshold.

For these reasons, the paper is deemed not acceptable to ICLR 23 in this form.


**Summary Of Ac-Reviewer Meeting:**

N/A